# Development of Hetero-Junction Silicon Solar Cells with Intrinsic Thin Layer: A Review

Nikolay Chuchvaga *, Kairat Zholdybayev, Kazybek Aimaganbetov, Sultan Zhantuarov and Abay Serikkanov

Institute of Physics and Technology, Satbayev University, Almaty A25A1G8, Kazakhstan
* Correspondence: chuchvaga@sci.kz or nikolay.chuchvaga@gmail.com

**Abstract:** This paper presents the history of the development of heterojunction silicon solar cells from the first studies of the amorphous silicon/crystalline silicon junction to the creation of HJT solar cells with novel structure and contact grid designs. In addition to explanation of the current advances in the field of research of this type of solar cells, the purpose of this paper is to show possible ways to improve the structure of the amorphous silicon/crystalline silicon-based solar cells for further improvement of the optical and electrical parameters of the devices by using of numerical simulation method and current hypotheses. This paper briefly describes the history, beginning from the first studies of and research of HJT-structure solar cells. It raises questions about the advantages and existing problems of optimization of HJT solar cells. The authors of this paper are proposing further ways of design development of HJT solar cells.

**Keywords:** HIT; HJT; solar cell; quantum wall; modeling; review



## 1. Introduction

The current challenges associated with the general growth of the world's population, political, economic and environmental circumstances put before humanity the search for new solutions in the field of production, processing and consumption of world resources. One of the key solutions to this problem is the development of green energy. The concept of "green energy" means energy resources that do not harm the environment and are, therefore, meant as environmentally friendly resources. The development of this industry is associated with the development of more highly efficient technologies for energy production and energy consumption of resources, the search for new materials, the training of specialists and the creation of infrastructure and jobs, as well as the creation of favorable conditions for the development of the green energy market.

One of the brightest and rapidly developing representatives is solar energy. This type of energy is associated with the conversion of solar energy, both for heat generation and with the use of semiconductor systems, for direct conversion of solar radiation into electricity. The main advantages of these systems are their efficiency, noiselessness, environmental friendliness and the ability to work in cloudy conditions and even in the rain. For the development of solar energy and its implementation in everyday life, there are tasks to increase efficiency, scale up production, search for new materials and, most importantly, reduce the cost of electricity produced. One of the most interesting materials capable of solving the presented problems is silicon. This is primarily due to the fact that silicon is the second most abundant chemical element on Earth. Despite the fact that silicon technology is the most traditional representative of solar energy, there are further technological ways to optimize the use of this material. One of the most prominent representatives of silicon technologies are silicon solar cells based on HIT and PERC technology.

In recent years, HIT structure solar cells (heterojunction with thin intrinsic layer) or, as it is also called—HJT—have gained great popularity. Such a big interest in this design is related to high efficiency of devices, low-temperature production technology (up to

200 degrees Celsius), as well as low degradation of photovoltaic structure properties. In this article, we would like to show the world achievements in the development of HJT technology, as well as speculate what engineering and technical perspectives await it.

## 2. Review

### 2.1. History

The technology of heterojunction silicon solar cells, also known as HJT solar cells (heterojunction technology), combines the advantages of crystalline and amorphous silicon, demonstrating the ability to achieve high efficiency of solar energy conversion when using less silicon and lower manufacturing temperatures that do not exceeding 200–250 °C compared to traditional diffusion technologies [1]. The first HJT solar cells were developed in the 1990s by Sanyo Company with an efficiency of 12% [2]. Since then, HJT technology has evolved, reaching new heights in terms of efficiency every year [3–5]. This technology has become one of the most promising for use in large-scale ground-based photovoltaics in a relatively short period, demonstrating record-breaking efficiency values of over 26% for all silicon technologies [6].

According to the calculations made by William Shockley and Hans-Joachim Quiesser, the theoretical efficiency limit for a semiconductor structure with one p–n junction is 33.7% [7]. Then, according to estimations presented on the work [8], it was shown that for silicon based solar cells, the maximum efficiency limit is 29.4%. The above-presented research works have set a goal for many engineers and scientists to strive for in silicon photovoltaics. Today, one of the fastest growing photovoltaic cell technologies is HJT technology.

Today the most popular on the market are solar cells based on crystalline silicon, in which a p–n junction is created by diffusion. This type of solar cell has advantages such as relatively high efficiency; it is also technologically well established. At the same time, this technology has big disadvantage of high temperatures up to 1000 °C; as a consequence, there is a high manufacturing cost. Crystalline silicon wafers of such solar cells must meet the requirements for high purity of the material. Together, all these parameters ensure high efficiency of solar cells. Amorphous silicon is a material that is perfectly suited as a cheaper alternative to crystalline silicon. Nevertheless, the efficiency of laboratory solar cells based on amorphous silicon does not exceed 14% [9].

In the first design version of these solar cells, the heterojunction was formed by using the flat n-type crystalline silicon wafer with a thin layer of p-type amorphous hydrogenated silicon (a-Si:H) deposited on its surface [2].

The efficiency of this structure reached 12.3%. This maximum value of efficiency was obtained for the thickness of the amorphous layer of the order of 100 Å (10 nm). It was also found that the introduction of intrinsic conductivity a-Si:H thin film between the n- and p-type silicon layers can reduce the density of surface defects at the interface between crystalline and amorphous silicon, which is reflected in simultaneously increasing the open circuit voltage, short-circuit current and solar cell fill factor. In this case, the efficiency of such a structure increased up to 14.8%, while the optimal thickness of the intrinsic amorphous silicon layer turned out to be 60–70 Å. Another innovation made by Sanyo was the creation of textured surface of the crystalline silicon wafer for more efficient absorption of light, as well as for the ability to absorb incident light on the wafer at low angles. Texturing conditions were not specified, but it was mentioned that silicon crystalline wafer was treated to hydrogen plasma before deposition of a-Si:H. In addition, a-Si:H layer of n-type conductivity was deposited on the back side of the silicon wafer to create an electric field on the back side of the solar cell. As a result, the efficiency of the solar cell reached a value of 18.1% for a 1 cm$^2$ sample.

Further improvement of the HJT technology was aimed at improving passivation of crystalline silicon surface by optimizing the amorphous silicon deposition conditions [10]. In 2009, Tsunomura and colleagues reported the creation of a high-efficiency HJT solar cell with dimension of 100.5 cm$^2$ [11]. The main factors used for the improvement were opti-

mization of the characteristics of the amorphous silicon–crystalline silicon heterojunction, reducing the thickness of the fingers and contact grid bars and reducing optical losses. In the fall of 2009, Sanyo presented a HJT-structure solar cell with silicon wafer thickness of 98 μm and an area of 100.3 cm$^2$ [12]. In early 2014, Panasonic achieved record efficiency of HJT cells by using a high-quality monocrystalline silicon wafer [13].

The essence of heterojunction solar cells is the formation of p–n junctions from materials with different values of the band gap. One of the main features of heterojunction silicon solar cells is passivation with a wide-gap semiconductor layer between the ohmic contacts and the active elements of the structure, which creates a high voltage when current flows through it; the voltage must be high enough to reduce the probability of recombination [14,15]. Silicon-based heterojunction semiconductor devices have a similar structure to metal-dielectric-semiconductor (MDS) devices based on the tunneling of charge carriers through a dielectric layer [16]. All this does not exclude the diffusion transport of charge carriers, which can play a major role in the current flow through the structure [17]. According to the information mentioned above, we can say that hydrogenated thin layers of amorphous silicon (a-Si:H) with thickness of a few nanometers are perfectly suitable as a buffer layer for semiconductor silicon multilayer solar cells. The bandgap width of amorphous silicon is larger than the bandgap of monocrystalline silicon, and it can easily be doped to achieve n- or p-type conductivity, which makes this material as suitable candidate for the role of buffer layer [18].

In 1974, V. Fuss conducted the first studies of a-Si/c-Si crystalline-silicon–amorphous-silicon heterojunction [19]. The first studies of passivation of crystalline silicon surface by amorphous silicon were shown in [20]. In 1983, the research group, in their work [21], presented a tandem-type solar cell with a-Si/c-Si structure. Sanyo (Japan) started introducing heterojunction solar cells with a-Si/c-Si structure of such structure in the 1980s. The manufactured devices consisted of n-type silicon wafers and emitters made of p-type conductivity amorphous silicon doped with boron. These solar cells had an efficiency of about 12%. Such small values of efficiency were due to the small value of fill factor caused by the high values of parasitic current [22]. However, one of the most significant steps made for creation of new high-efficiency HJT structure solar cells was the idea of the using of the intrinsic conductivity a-Si:H as a buffer layer between the doped emitter and the silicon wafer which led to reduction of the dangling bond densities and densities of defects at the interface. This structure is called a heterojunction with intrinsic thin layer (HIT). The efficiency of the first elements of the HIT structure was 14.8% ([2], p. 3521). In the work ([2], p. 3522), a created similar heterojunction structure solar cell was presented. The solar cell held, at that time, a record efficiency of 18%. The novelty of this work was the application of intrinsic amorphous layer on the back side of the solar cell.

The authors of this work have clearly demonstrated what design a modern heterojunction solar cell should have; namely, it should have buffer passivation layers on both sides of the silicon wafer, as well as an integrated thin layer of intrinsic conductivity between the doped emitter layer and the crystalline wafer.

### 2.2. Degradation and Passivation of Surfaces Defects

An integrated intrinsic conductivity a-Si thin layer plays the role of passivation of the surface states of the crystalline silicon. It is applied between the surface of the crystalline silicon and the doped a-Si layer. The doped amorphous layers passivate the surface of the crystalline substrate by themselves, but as shown in [23], intrinsic conductivity amorphous layers passivate the substrate surface much better. It is worth noting that this dependence is explained by changes in the value of the Fermi level [24]. Such a change leads to a decrease in defect formation energy. Consequently, an increase in doping concentration can lead to a higher defect density. Studies on the relationship between doping levels and defect formation are highlighted in [25,26]. One of the effects that negatively affects the efficiency of photocells is the Staebler–Wronski effect [27]. It is known that prolonged illumination of amorphous hydrogenated silicon (a-Si:H) leads to changes in the conductivity of the

structure, including photoconductivity. Many studies have been conducted on the subject of the Staebler–Wronski effect [28–32]. Many of these works have shown that the change in photoconductivity of the structure under prolonged illumination is associated with a shift of the Fermi level. The shift is due to increasing defects associated with broken bonds of the amorphous material. However, a model of recombination in a-Si:H p-type was proposed in work [33], which corresponds well to the experimental data. According to the proposed model, for a-Si:H p-type films having the same parameters of valence band tail states, the photoconductivity should not depend on the doping level and total concentration of defects of the broken bond type. The proposed recombination model fully explains the effect of long-term illumination on the temperature dependences of photoconductivity of the a-Si:H p-type.

Therefore, one of the important areas of research in the field of heterostructured solar cells is the study of photoinduced degradation processes. In paper [34], the object of the study was solar cells with a structure of $\alpha$-Si:H/$\mu$c-Si:H, fabricated by the modified technology of OerlikonSolarLtd (Switzerland). Studies of photoinduced degradation at temperatures of 298, 328 and 353 K were conducted. The experimental data obtained were used to estimate the magnitude of the change in the concentration of free (broken) bonds after saturation photoinduced degradation from temperature. The presented model assumes that a large fraction of hydrogen is in the semiconductor in the form of metastable Si–H–H–Si complexes. The formation of free bonds occurs due to the rupture of the weak Si–Si bond in close proximity to the metastable Si–H–H–Si complex. After the rupture of the weak Si–Si bond, the metastable complex breaks and a pair of free bonds and a pair of hydrogenated Si–H bonds are formed. At high temperatures (>345 K), the defects are annealed.

Against the background of the studied materials on the Staebler–Wronski effect, a very interesting work appears in which the abnormal Staebler–Wronski effect is seen in the amorphous silicon [35]. A description of the effect, its justification and the mechanism of this effect were given. The essence of the observed effect is an increase in the photoconductivity of a-Si:H thin films after taking a light bath. The authors showed that such behavior of the material should be well described by the Debye model and Williams–Watts model. Applying this anomalous effect in SHJ solar cells, the authors achieved a power conversion efficiency (PCE) of 25.18% (26.05% on a given area) with an FF of 85.42% on 244.63 cm$^2$ plates. This PCE is one of the highest values for silicon solar cells with a top/back contact on the total area. The FF value reaches 98.30% of its Shockley–Quisser limit.

### 2.3. Features of HIT Solar Cells Production

The use of amorphous silicon makes it possible to carry out technological processes for the production of solar cells at low temperatures not exceeding 200 °C, which allows the use of a smaller number of solar cells manufacturing stages and reduces its cost. The process of deposition of amorphous silicon on a crystalline wafer occurs by the method of plasma-chemical vapor deposition (PECVD-plasma enhanced chemical vapor deposition). For creation of the p–n junction, the material is doped by adding another gas agent (C$_3$H$_9$B for p-type conductivity and PH$_3$ for n-type conductivity). The intrinsic layer of amorphous silicon is deposited from silane (SiH$_4$) with a large proportion of hydrogen (H$_2$) for hydronization [36]. From the opposite side of the emitter, a thin layer of intrinsic conductivity is similarly deposited and a layer of amorphous silicon of the same type of conductivity as the substrate is deposited. This combination forms the so-called BSF-layer (-back surface field). The BSF is formed in order to prevent minor charge carriers from reaching the external circuit and thereby creating parasitic reverse currents. At the next stage, a layer of TCO (transparent conductive oxide) with a low resistance of less than 100 Ohm/square is formed on top of the finished "sandwich" by magnetron sputtering. Mostly, the material used as TCO in HJT solar cells is indium tin oxide (ITO). After the deposition of ITO layers, metal contacts are applied by screen printing method: (a) on the both sides of the double-sided solar cells; (b) on the front side for single-sided solar cells. For a single-sided

solar cell, the back side metal contact is deposited by magnetron sputtering immediately after applying the TCO layer. Further, the resulting cell is connected with other similar solar cells into a module and laminated.

From the above-mentioned information, we can conclude that thin films of amorphous hydrogenated silicon are well suited for passivation of the surface states of a crystalline silicon substrate [20]. The a-Si:H film is deposited by the PECVD method from silane ($SiH_4$) diluted in hydrogen. The plasma discharge frequency is typically 13.56 MHz [37]. However, it should be noted that different equipment may have different optimal deposition parameters [38].

Passivation of the crystalline wafer surface by the deposition of amorphous silicon occurs by closing of dangling bonds with hydrogen, which leads to a decrease in the density of surface states ([37], p. 4440). Presumably, the defect that is responsible for surface recombination is a dangling bond on the silicon surface. These defects can be neutral, negative or positive. Such a surface recombination model was experimentally confirmed for a-Si:H/c-Si structures [39]. The described passivation of the wafers should rid the surface of defects of this kind, which lead to active surface recombination.

The doped layers in the HJT structure play the role of an n/p-type emitter or p+/n+-type BSF layer. These films are deposited by the PECVD method in the same equipment as the layers of intrinsic amorphous silicon. Usually, different chambers are used for each type of film, but all layers can be deposited in one chamber if the chamber is cleaned and conditioned before deposition. The agent gases used for creation the conduction layer are trimethyl boron (TMB) or biborane ($B_2H_6$) for p-type films; and phosphine ($PH_3$) for n-type films. All process gases in the working chambers are highly diluted in hydrogen.

An increase in the lifetime of charge carriers is achieved by passivation of surface states by deposition of thin amorphous layers which allows charges to pass without recombination through all layers of the structure to external metal contacts. High lifetime values (on the order of several thousand microseconds) and the ability to deposit contacts directly over the conducting oxide, avoiding contact between the metal and the emitter, eliminate recombination losses at the interface between the metal and doped amorphous silicon and make it possible to achieve high open-circuit voltage (Uoc). However, since the lifetime of charge carriers in amorphous films is many times shorter than in crystalline silicon, this has a negative effect on the conductivity of the structure. Since visible light is absorbed in the crystalline wafer, the negative effect is felt only from the front side of the solar cell which is expressed by the drop in short-circuit current (Isc) [40].

HJT structure have a sandwich design consisting of five main layers: p-type amorphous silicon layer, a built-in intrinsic amorphous silicon layer, n-type single-crystal silicon layer, a rear amorphous intrinsic layer, and a heavily doped amorphous silicon layer of n-type conductivity (Figure 1). A p-type conductivity layer forms a p–n heterojunction with a layer of crystalline silicon. A heavily doped n-type layer of amorphous silicon creates a back blocking field, which creates an energy barrier for minor charge carriers. Embedded thin amorphous layers of intrinsic conductivity provide passivation of the surface states of the crystalline silicon substrate [41–44].

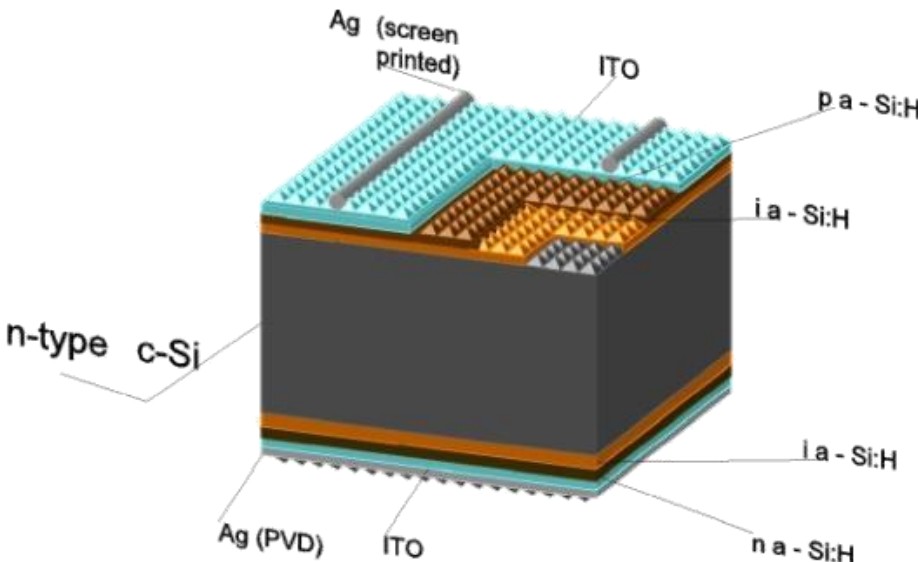

**Figure 1.** Schematic view of the HJT solar cell [45].

*2.4. Design and Parameters of the HIT Structure*

One of the first research works showed the importance of determining the most efficient parameters of HJT solar cells. The authors showed the influence of the thickness of the emitter and intrinsic built-in layer of the photocell structure [2]. The authors showed that the solar cell efficiency starts to increase after the deposition of intrinsic amorphous silicon layer. The dependence curve increases to a maximum and then starts to decline. This indicates the importance of the correct selection of the parameters of the solar cell.

In order to achieve a device efficiency that is as high as possible, it is necessary to determine what the main output characteristics of device depend on and find the optimal parameters that can be achieved during the growth of heterostructure. As shown by the authors of some works [46,47], one of the most important parameters is the quality of the crystalline silicon layer in the HJT structure. The density of surface defects $D_{it}$, as well as the density of bulk defects (in p-type crystalline silicon, such defects are caused by oxygen inclusions $D_{od}$) are important parameters; the values of these parameters depend on the method and quality of material growth and affect the output characteristics of solar cells. Considering the following structure of solar cell: TCO/a-Si:H(n)/a-Si:H(i)/c-Si(p)/ Al-BSF/Al, the authors showed that for the minimum efficiency loss of the solar cell $D_{it}$ should not exceed $10^{11}$ cm$^{-2}$, and $D_{od}$ should not exceed $10^9$ cm$^{-2}$. Similar results were shown in the work in which the TCO/a-Si:H(n)/c-Si(p)/Al/structure was modeled. According to the authors, the optimal value of $D_{it}$ should be no more than $10^{10}$ cm$^{-2}$.

One of the parameters controlled during the fabrication of solar cell is the thickness of one or another layer of the structure ([2], pp. 3519–3520). According to the researcher works, the thickness of the crystalline silicon layer in the HJT device should be 50–300 microns, depending on its structure. The optimal thicknesses of the amorphous emitter and intrinsic layer are 5–10 nm and 3–5 nm, respectively [47–50].

One of the most important factors affecting the performance of HJT cell is the material interface between the TCO layer and amorphous material, and between crystalline and amorphous silicon. The studies of these factors are described in some theoretical works devoted to the modeling of solar cells ([48], p. 168).

According to the research works, the optimal energy for breaking the bands in the c-Si/a-Si heterojunction is $\Delta EV = 0.37$ eV and $\Delta Ec = 0.25$ eV, since a larger potential barrier for the main charge carriers leads to a drop in the short circuit current and, consequently, to a drop in the efficiency of the solar cell. It is also shown that when the work function of the TCO layer is WTCO = 5.1–5.2 eV, an increase in the output characteristics is observed; at lower values, a decrease occurs. This is due to the charge distribution in the depletion

region, which affects the quasi-Fermi levels in these regions, which leads to a change in the band diagram and a decrease in the efficiency of the solar cell.

An important controllable input parameter of a solar cell that affects its output characteristics is the concentration of charge carriers in each layer of the structure. According to the works ([50], p. 0111), the concentration of the main charge carriers of the p-type emitter layer in the HJT solar cell should be at least $2 \times 10^{20}$ cm$^{-3}$. Unfortunately, there are not so many works on the study of the influence of the concentration of charge carriers of the layers on a solar cell operation.

## 3. Modeling and Optimization of Structure

Recently, quite a lot of work has been devoted to research using computer modeling tools. For example, in paper [45], taking into account the experimental data, as well as using the AFORS-HET program, the optimal design of the HIT photocell structure was calculated (Table 1).

**Table 1.** Optimized parameters of the HIT solar cells simulated using the AFORS-HET tool [45].

| | | | |
|---|---|---|---|
| *dp-a-Si* | 7 nm | *Na, p-a-Si* | $1 \times 10^{20}$ cm$^{-3}$ |
| *dn-c-Si* | 260 μm | *Nd, n-c-Si* | $2 \times 10^{17}$ cm$^{-3}$ |
| *dn-a-Si* | 20 nm | *Nd, n-a-Si* | $1 \times 10^{20}$ cm$^{-3}$ |
| *di-a-Si*, front | 5–7 nm | *di-a-Si*, rear | 5 nm |

The authors also showed that the physics of the heterojunction must be taken into account to advise the validity of the calculations. For this purpose, the authors of the paper used the Anderson heterojunction model [51].

Based on the laboratory optimization of the HIT cell structure, optimal parameters of a HIT-structured solar cell were elucidated for the given wafer quality, and an optimized solar cell was fabricated (Table 2) with the following output characteristics: $J_{SC}$ = 37.51 mA·cm$^{-2}$; $V_{OC}$ = 713 mV; FF = 77.18%; PCE = 20.64%. The simulated values for comparison were as follows: JSC = 30.33 mA·cm$^{-2}$; VOC = 781.6 mV; FF = 85.44%; PCE = 20.26%.

**Table 2.** Final optimized parameters of HIT solar cells [45].

| | | | |
|---|---|---|---|
| $d_{p\text{-a-Si}}$ | 10 nm | $N_{a,p\text{-a-Si}}$ | $3.59 \times 10^{20}$ cm$^{-3}$ |
| $d_{n\text{-c-Si}}$ | 170 μm | $N_{d,n\text{-c-Si}}$ | $2.15 \times 10^{17}$ cm$^{-3}$ |
| $d_{n\text{-a-Si}}$ | 20 nm | $N_{d,n\text{-a-Si}}$ | $1 \times 10^{20}$ cm$^{-3}$ |
| $d_{i\text{-a-Si, front}}$ | 7 nm | $d_{i\text{-a-Si, rear}}$ | 9 nm |

In paper [52], the authors presented an analytical study of the effect of light trapping and multilayer anti-reflective coating (ARC) on the electrical characteristics of n(a-Si:H)/i(aSi:H)/p(c-Si)/p+(C-Si) heterojunction solar cells with an internal thin layer (SHJ). Due to the improved light trapping capability provided by the optimized triangular texture morphology, a conversion efficiency of 20.06% was achieved. This result was compared with similar c-Si based structures with a planar structure and a single ARC layer (Figure 2).

Another example of using the AFORS-HET program is paper [53]. In this work, to simulate a solar cell with AZO/Si and TiO$_2$/Si heterojunction, the AFORS-HET automatic simulation program is used, in which the ultra-thin AZO layer and TiO$_2$ layer act as n-type layer, and the crystalline silicon wafer of p-type (p-cSi) acts as an absorbing p-type layer. In this paper [53], texturing is performed at different texture angles and the performance of a simulated AZO/Si heterojunction solar cell is optimized. The role of surface texturing at different angles was studied and maximum efficiencies of 17% and 17.5% were obtained for silicon HJ solar cells based on AZO and TiO$_2$ layers, respectively. A linear increase in the efficiency as a function of the texturing angle is observed, where the efficiency is 14.21% for the AZO layer and 14.45% for silicon HJ solar cells based on the TiO$_2$ layer when the surface is pyramidally textured. Optimizing the p-cSi thickness at 70 μm

and removing the amorphous silicon inner layer can be very cost effective for producing AZO/Si heterojunction solar cells on an industrial scale for commercial production, as deposition of the a-Si i-layer and other similar HJ-based solar cells with an inner layer requires additional processing.

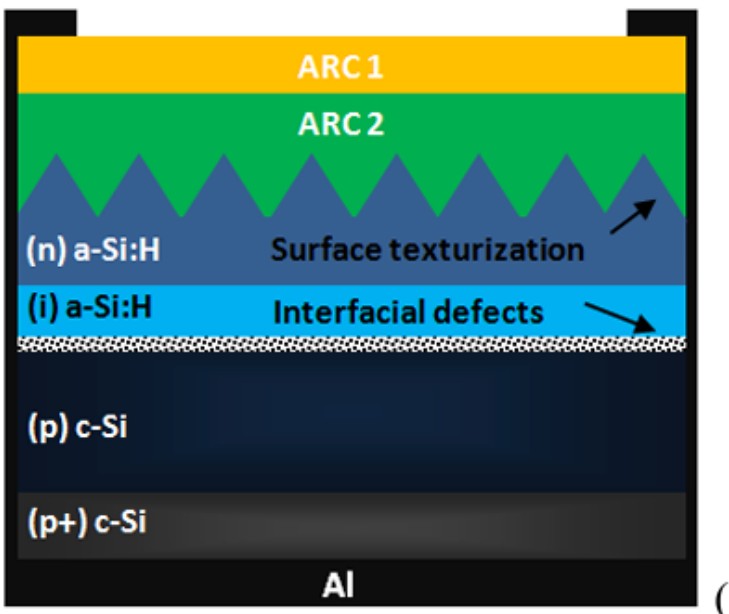

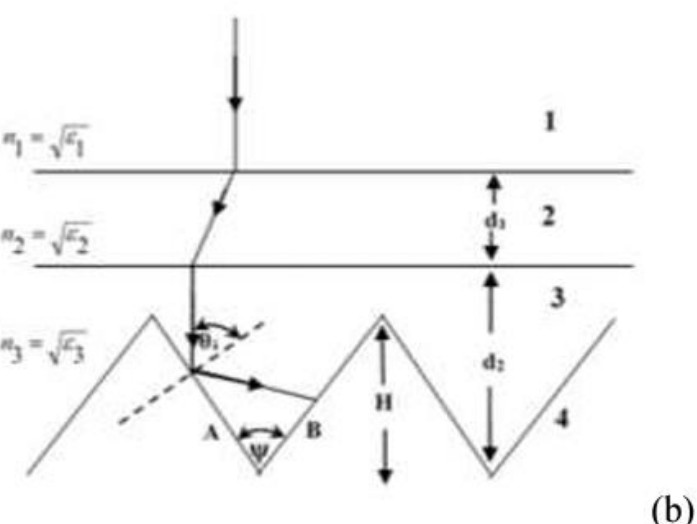

**Figure 2.** A schematic representation of the double ARC layer and texture morphology of the cell [52]. (**a**) Cross-sectional view of the proposed SHJ solar cell. (**b**) Schematic representation of the double ARC layer and triangular texture morphology.

To the disadvantage of this work and other works related to the use of software packages, I would attribute the exclusive subjectivity of the conclusions based on the analysis of the computer program model without further experimental research. However, even research at this level raises interesting topics and introduces new ideas, which is undoubtedly useful for world science.

To demonstrate the diversity and choice of modeling tools, we offer the review paper [54]. In this paper, the HIT solar cell is simulated by Amorphous Semiconductor Device Modeling Program (ASDMP). The influence of the physical and geometrical parameters of

the emitter on the performance of the element, the phenomena of conduction and recombination have been studied. It is shown that when choosing the thickness of the emitter, a compromise must be found: It must not be too thin so it has a sufficient electric field at the p-a-Si:H/n-c-Si junction; and it must not be too thick such that the maximum photons are transferred to the active layer and so that the diffusion length of charge carriers was at least equal to the thickness of the emitter. With a p-Si:H layer thickness of 155 Å, the best values of the cell parameters were achieved, while the efficiency was 18.25%. The performance of the investigated HIT cell is improved by increasing the doping density of the emitter layer of hydrogenated amorphous p-type silicon (p-a-Si:H). Despite the deterioration of material characteristics as a result of doping, the recombination rate decreases on the active layer (n-c-Si) due to the higher electric field strength.

Another interesting paper that talks about a new method [55]. A new method is presented for extracting seven parameters of a two-diode solar cell model using the current-voltage (I–V) characteristics under illumination and in the dark. The algorithm consists of two subroutines that are executed in turn to adjust all cell parameters in an iterative process. The proposed parameter extraction algorithm uses the I–V characteristics in light and darkness so it can predict the MPP and, at the same time, give a better idea of the physical structure of the solar cell, since the accuracy of performance prediction in both conditions is improved. In particular, the proposed parameter extraction algorithm is divided into two subroutines: IIVf (I–V fitting in lighting) and DIVf (I–V fitting in dark), which are executed alternately until a good match is achieved in both conditions. The proposed method can be used to extract the parameters of any photovoltaic device, either commercial or fabricated in research laboratories, as long as their operation relies on the p–n junction.

The novelty of the method should be attributed to the advantages of work [55]. The possibility of using it in conjunction with electrical measuring techniques. The authors of the work also showed the efficiency of the method experimentally.

## 4. Discussion and Possible Solutions

Finally, we come to the question: "What else can be done to improve the efficiency of the HJT-solar cells?". We are interested in engineering solutions in the design of the structure, and not in improving the quality of semiconductor and metallic materials.

Samples produced in laboratories achieved efficiency up to 26.7% [56]. These record results are the result of optimization and development of the HJT-structure. There are three directions which can be performed for the optimization of the HJT-structure: improving material quality and production methods, optimizing the internal parameters of the device (thickness, concentration of layers) and changing the design.

### 4.1. Contact Design

For example, several types of metal contact grid designs are developed: "Busbar", "SmartWire" and "IBC". Solar cells with BUSBAR contact design are metallized with thin rectangular strips printed mostly only on the front and rarely, additionally, on the back of the cell. These metal contacts are called "busbars" and have an important purpose: they carry the direct current generated by the solar cell. Perpendicular to the "busbars" are thinner metal "fingers" that collect the generated current and deliver it to the "busbars". The cost of metallization for the production of heterojunction solar cells has been significantly reduced by applying the "SmartWire" contact grid design [57]. This design replaces the busbars with a lot of thin wires. The soldering of such contacts with the "fingers" contacts occurs directly during the lamination of the solar cell.

The third contact design is "IBC", the structure of which is shown in Figure 3. The use of such a design of contacts allows increasing the exposure to sunlight solar cell surface area, which has a positive effect on the short circuit current of the solar cell.

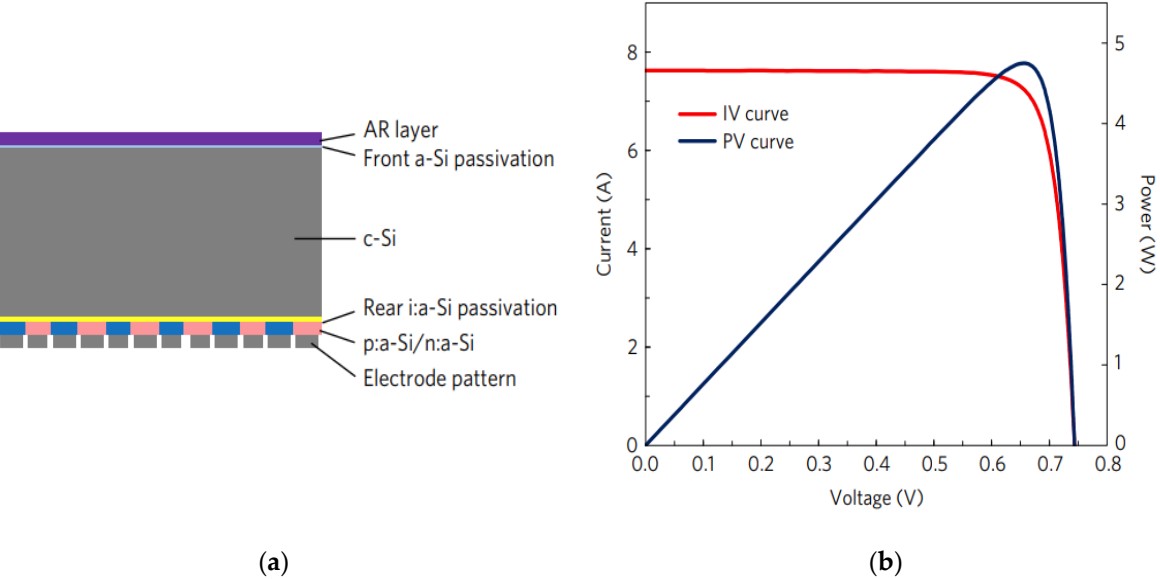

**(a)**                                                                                                                                                    **(b)**

**Figure 3.** Data presented from research work [6]: (**a**) HJT solar cell structure with integrated rear contacts; (**b**) IV-curve of the presented structure.

The highest efficiency values for HJT solar cells are shown by the IBC contact design [6,58]. This assertion can be verified by tracing some works devoted to Sokar Cells Tables [3–5,9,56]. In this architecture, by integrating both P+ and N+ HJT contacts on the back side, good optical and passivation properties can be obtained on the front side of the cell that receives light. As shown in Figure 3a, the amorphous Si p-type layer (p:a-Si) and the amorphous Si n-type layer (n:a-Si) are cut to collect holes and electrons, respectively. The face is coated with an a-Si passivation layer and a dielectric anti-reflection (AR) layer. The incident light generates carriers that are collected by the patterned a-Si layers and electrodes [6].

*4.2. Internal Layer Modification and Defect Engineering*

The results achieved during the studies have demonstrated the effectiveness of intrinsic amorphous silicon in the surface passivation of crystalline silicon wafers. The built-in layer of amorphous silicon is used in the HJT solar cell and aimed to passivate heterojunction defects. However, the presence of this layer also leads to undesirable consequences for the solar cells; namely, due to the qualities of the amorphous material itself, a drop in the diffusion length of charge carriers in the area is observed and, as a result, reduces the efficiency of the solar cell. One of the possibilities to further improve the efficiency of a heterojunction silicon solar cell is to increase the diffusion path of charge carriers in the heterojunction region [59,60]. The research work [61] presents a model of a solar cell with a very thick i-a-Si-layer, which, according to the authors, increases the generation of electron–hole pairs and prevents overheating of the electronic device. In the work [45,62], the experimental dependences of the output electrical characteristics of the HJT solar cell on the parameters of the internal layers is shown. Based on the research works, as well as to the work [63], it can be concluded that in HJT structure solar cells, an increase in the thickness of the built-in amorphous layer, the efficiency of the solar cell increases, but up to a certain maximum point. The increase in output characteristics is due to an increase in the lifetime of minority charge carriers in the crystalline silicon wafer. This behavior of the system can be explained by better passivation of the surface states on the silicon substrate.

The search and creation of an alternative semiconductor material will make it possible not only to use it in solar energy, but also in other areas of electronics where there is a need for such semiconductor films with high charge mobility and a relatively cheap production method.



The end result of the work is a film or heterojunction, an alternative to the "single-crystal silicon-hydrogenated amorphous silicon" junction. Currently, in a standard HJT solar cell, a photoelectric effect occurs in the p–i–n junction. In this technology, the i-layer performs the role of passivation of surface states on the surface of the crystalline substrate and serves as a buffer layer between p and n. However, with an increase in the geometric dimensions of the field that is created by the p-i-n junction, it is possible to increase the region of useful absorption of photons by the material. Since amorphous silicon is a rather defective material, with an increase in the thickness of its own layer, the efficiency of the solar cell decreases. The main hypothesis is that the pores on the crystalline wafer filled with a thick layer of amorphous silicon will play the role of areas that passivate the dangling bonds of the crystalline surface on themselves, and the places on the surface of crystalline silicon covered with a thin layer of amorphous film will be responsible for the current conductivity in structure. Thus, it will be possible to achieve good conductivity with better passivation of the structure. In addition, possible quantum-size effects, depending on the obtained structures, can affect the final conductivity of the structure. Testing this hypothesis and developing methods for creating such a structure are the priority tasks of our research team at the moment. Today, the work is not fully completed. Recipes for obtaining porous silicon of various topologies are being developed.

The next hypothesis that our research group is proposing is to test is the creation of domain structures with different effective masses of charge carriers or different conductivity. For example, we are proposing the use of composite material as semiconductor material. It is proposed to introduce metal nanocrystals into an amorphous material (which will serve as a matrix for a composite material), somehow "include" such nanocrystals in the structure of an amorphous semiconductor and obtain domains with changed charge carrier parameters in certain areas.

As an example of the successful application of such an approach, I would like to describe the work of scientists from Spain [64]. The authors made a HIT solar cell: in its design, a hyperdoped layer was applied to the surface of crystalline silicon-photovoltaic cell based on intermediate band (IB) semiconductors (IB solar cell, or IBSC) (Figure 4).

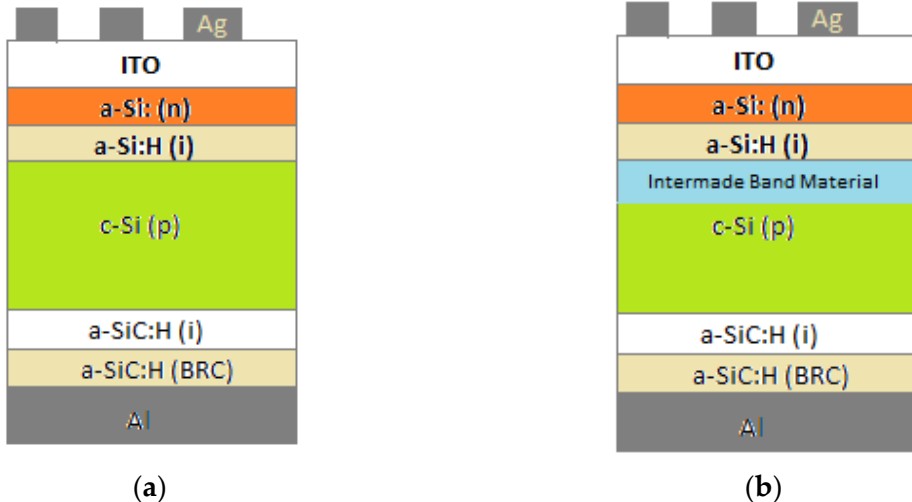

(**a**)  (**b**)

**Figure 4.** An example of structure of the reference HIT solar cell (**a**) and IBSC (**b**) used in [64].

In this study, the researchers created various IBSCs based on a hyperdoped silicon semiconductor with Ti and V. These cells have an efficiency of 2.6% and 1.9%, respectively, and exhibit a silicon subbandgap EQE. Analysis of J–V measurements showed three main conduction processes and series resistance. The main exponential mechanism observed in all temperature ranges in this study has an ideality coefficient of $1 < n1 < 2$ and an activation energy of Ea (Ti) = 0.55 eV and Ea (V) = 0.56 eV.

In this work [64], the authors failed to increase the efficiency of the HIT solar cell. However, at the same time, we believe that this direction—the direction of engineering and the study of the properties of heterojunctions—is worth moving towards.

### 4.3. Application of the Properties of Nanoscale Structures

Continuing the theme of changing the properties of the i-layer, let us consider the use of quantum wells in the solar cell structure.

In the work [65], a simulated situation is shown, in which hot electrons in a semiconductor lose their energy, falling on the "shelves" of quantum walls. The negative effect of Auger recombination of electrons on the conductivity of the structure is thereby reduced (Figure 5).

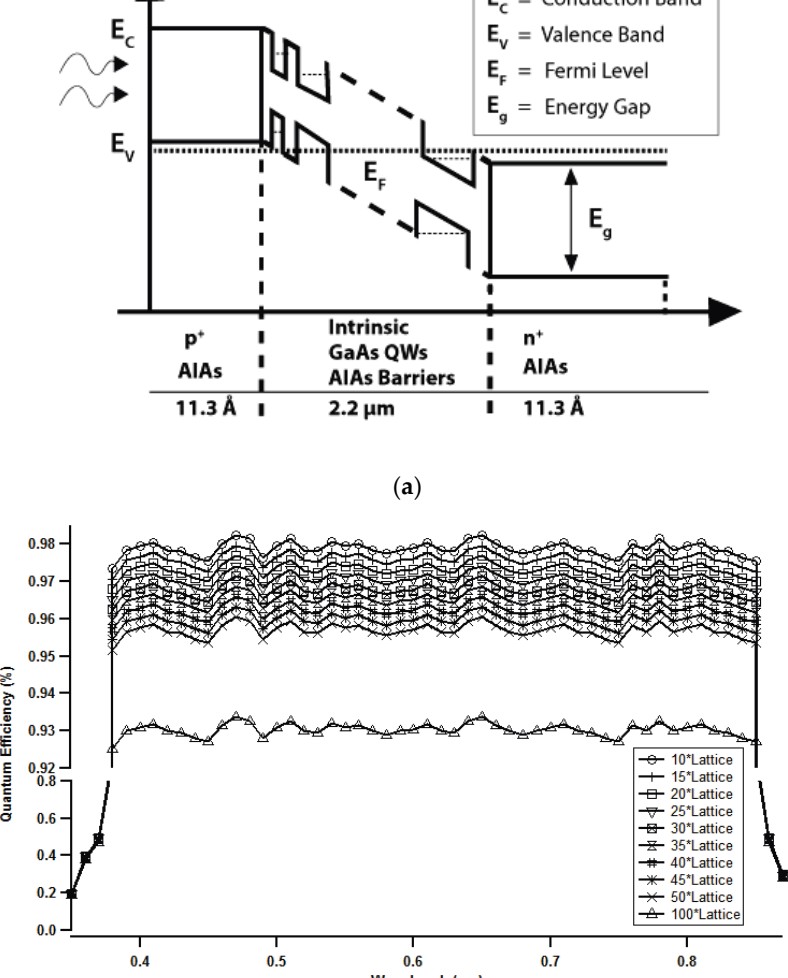

(**a**)

(**b**)

**Figure 5.** This is a figure data presented from [65] with the permission of AIP Publishing: (**a**) Energy diagram of a heterojunction solar cell with p–i–n QW; (**b**) QW efficiency versus wavelength for all 10 sets of QWs.

If we continue to consider quantum phenomena in a semiconductor, let us consider a superlattice of quantum wells. A superlattice is a solid-state structure in which, in addition to the periodic potential of the crystal lattice, there is an additional periodic potential: the period of which significantly exceeds the lattice constant.

For non-degenerate energy bands, we have:

$$-\frac{\left(\frac{h}{2\pi}\right)^2}{2m_{eff}}\nabla^2\psi(r) + \Delta(r)\psi(r) = \varepsilon\psi(r) \tag{1}$$

where $m_{eff}$ is the effective mass of an electron or hole.

It is shown that the energy spectrum of the superlattice has a band character, determined by the band number and the wave vector [66].

The qualitative properties of the energy structure of a superlattice (SL) are the same for different SLs. The spectrum $\varepsilon j(k)$ is a series of non-overlapping minibands. If the energy of the miniband is less than the maximum of the SL potential, then such minibands have a small width, determined by the tunneling transparency of the SL barriers; these minibands can be described in the tight coupling approximation:

$$\varepsilon j(k) = \varepsilon j - \Delta j \, \cos kd \tag{2}$$

where $\varepsilon j$ are the energy levels of a single well; $|\Delta j|$-width of $j$ mini-zones, which are determined by the parameters of the SL.

Figure 6 shows the presence of maxima in the I–V characteristic, which can result in the possibility of a negative differential resistance of the structure. This property of the structure with quantum wells can be used to determine the optimal values of the internal field of the p–n junction of the solar cell.

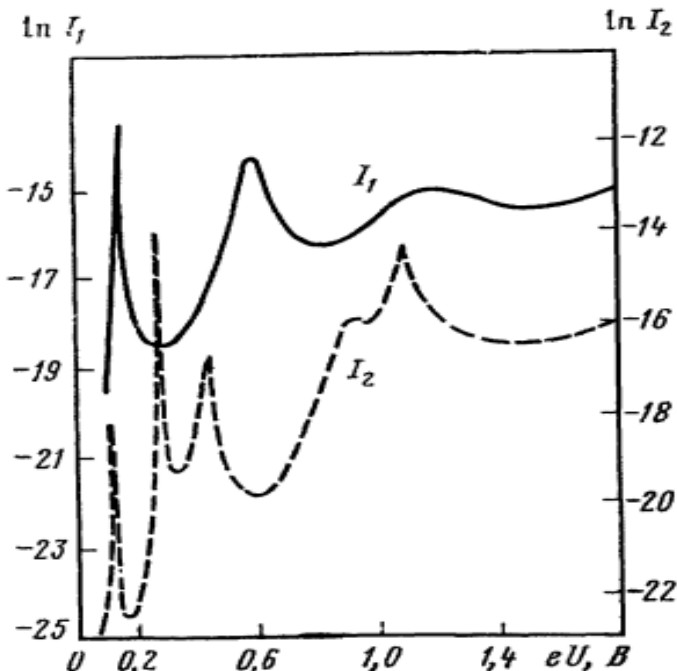

**Figure 6.** Dependence of the natural logarithm of the current passing along the SL on the voltage for structures with two and three barriers [66].

As it is well known, if the barriers between QWs are sufficiently transparent. That is, the tail of the wave function of an electron in one state of QWn intersects with the tail of the wave function of an electron in the state of QWn + 1. Then, the electron can tunnel to another state through the energy barrier. Moreover, if the energy levels of neighboring states have the same value, then the probability of tunneling increases since it does not imply a change in the electron energy. This suggests that if there is a superlattice inside the p–n junction, it is possible to increase the region of the i-layer of the p–i–n structure due to the tunneling current in the SL.

### 4.4. Modern Tendencies

In conclusion, we would like to give a graph of the development of solar cell technology from the well-known resource NREL. As you can see from Figure 7, heterojunction technology tends to grow steadily in efficiency, and it has quickly overtaken other technologies. Now, of the solar cell technologies that are designed for civilian applications, only the new perovskite [67] and organic [68] technologies are rapidly gaining the upper hand. However, these technologies have not yet perfected the production process; they have a number of shortcomings at this stage of development [69].

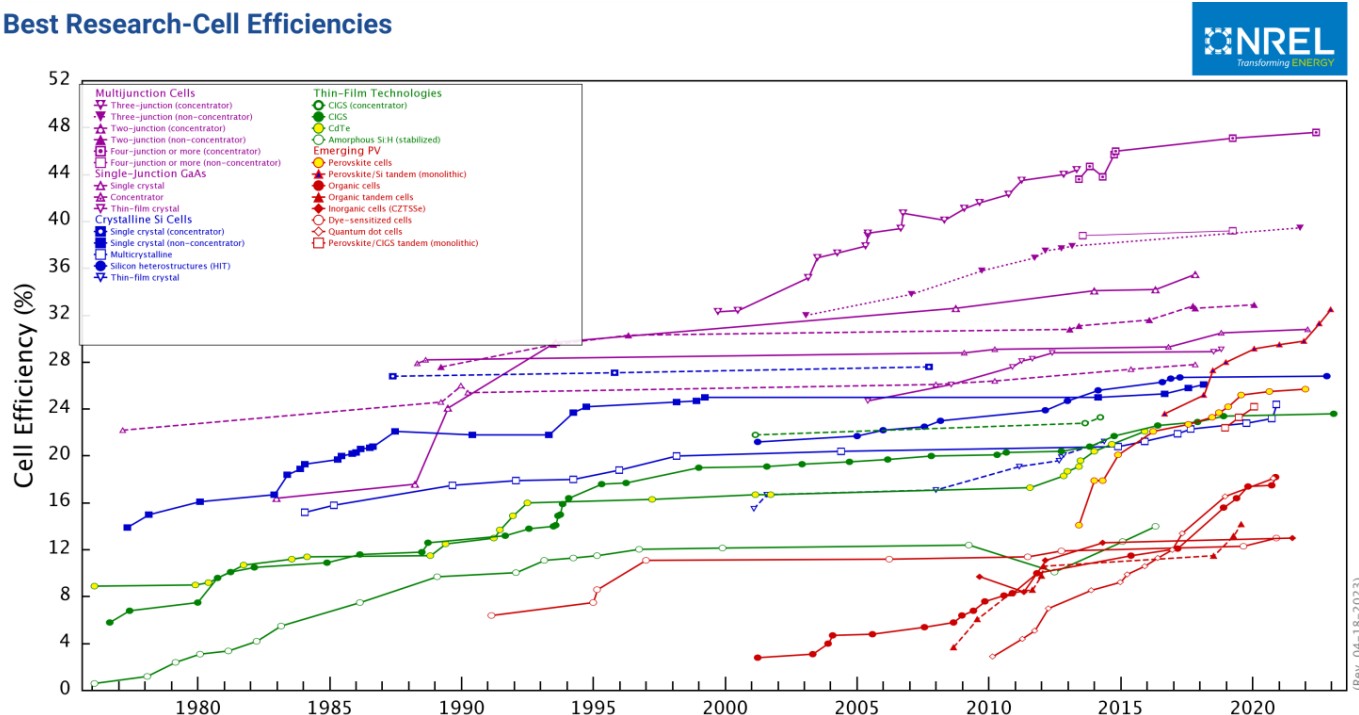

**Figure 7.** Best Research-Cell Efficiency Chart presented by NREL. The plot is courtesy of the National Renewable Energy Laboratory, Golden, CO [70].

Based on the above, it is not surprising that, at present, there are many works devoted to the tandem of solar cells based on the perovskite structure and other photocells [71,72].

Figure 8 shows variants of tandem solar cells based on silicon and perovskite. Module C is a variant of using the HIT photocell ([71], p. 1561). Figure 7 shows that the record of such a tandem photocell is 32.5% [73]. The connection options for tandem solar cells can be divided into several types. One popular connection type is where the top and bottom cells can independently contribute to the maximum power output, as the top and bottom cells are only connected optically without electrical connection, using the top solar cells as filters (Figure 9a). The other type is the two-terminal (2T) tandem solar cell (Figure 9b), which has the advantage of less parasitic absorption because it is a simple integrated type without an additional glass substrate and a thick transparent electrode for the top perovskite cell. However, it requires sophisticated technology such as process optimization and current matching technology [74]. More details about the possible options for tandems of perovskites and heterotransition silicon elements are found in review [75].

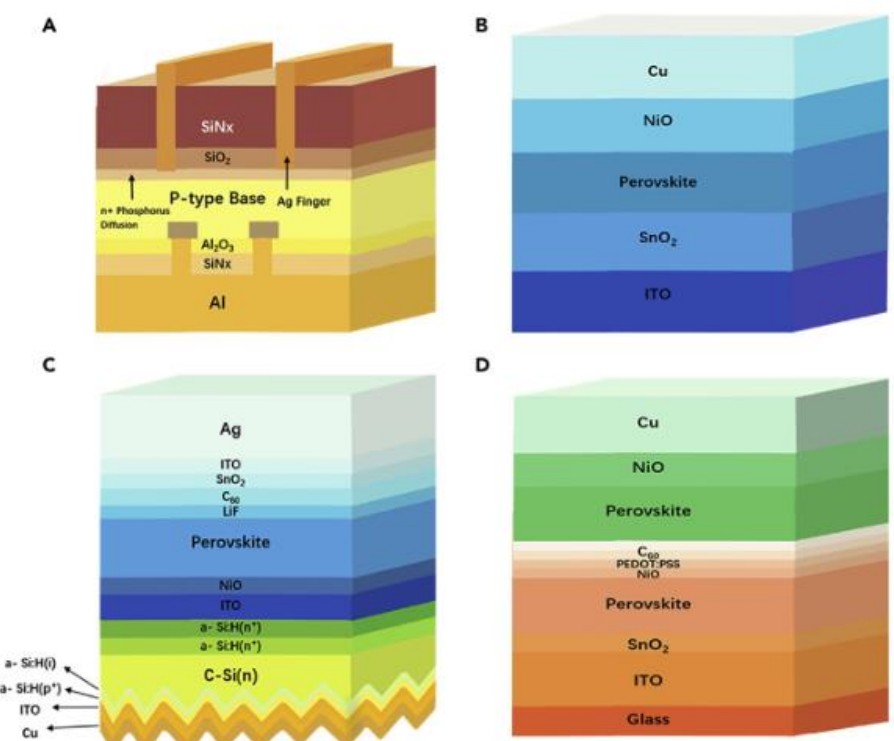

**Figure 8.** This is a figure data presented from ([71], p. 1561): Schematic diagram of solar cells. (**A**) is composed of traditional silicon cells, (**B**) is composed of planar perovskite cells, (**C**) is composed of silicon/perovskite tandem cells, (**D**) is composed of perovskite/perovskite tandem cells.

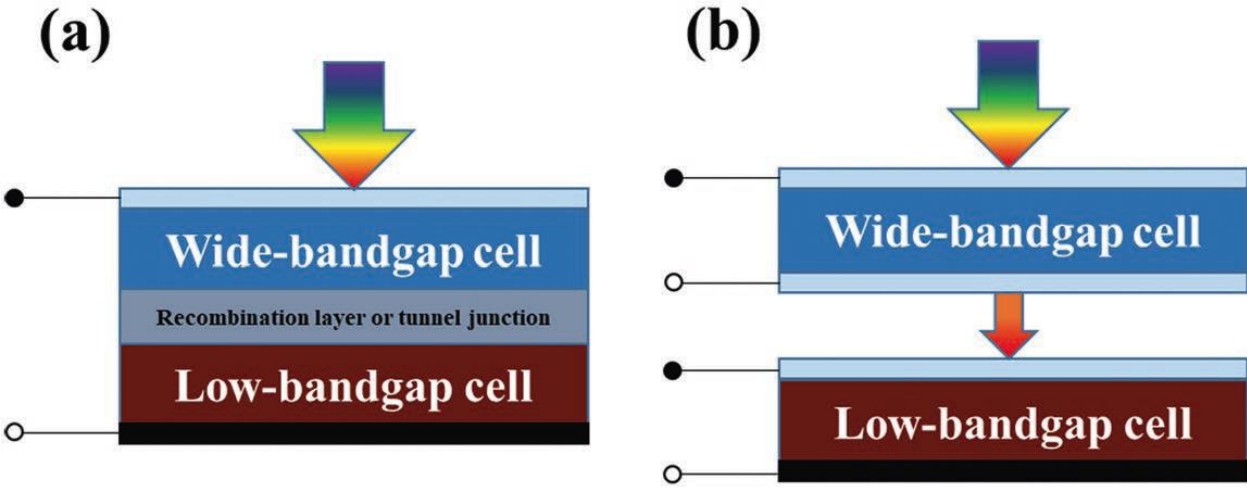

**Figure 9.** This is a figure data presented from [74]. Schematics of tandem architectures: (**a**) 4-terminal mechanically stacked (**b**) 2-terminal monolithically integrated [74].

### 4.5. Other Applications with Heterojunction Photocells

Recently, works related to the electrochemical etching of silicon in installations which use photocells have been appearing. In Ref. [76], the paper showed that a heterojunction photocell can be used as a photoanode and a photocathode. The current density in such a setup was 21.48 mA/cm$^2$. The conversion efficiency was 13.26%. Figure 10 shows a schematic of the cell.

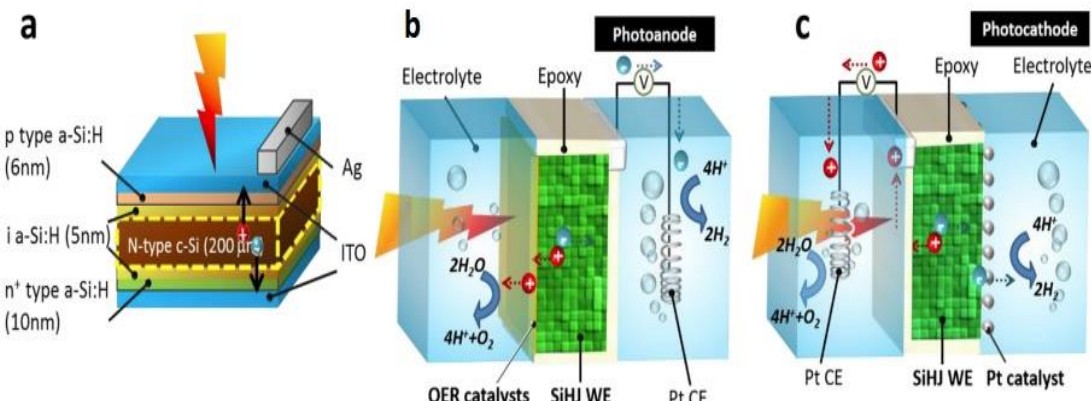

**Figure 10.** This is a figure data presented from [76]. (**a**) Schematic of the SiHJ solar cell and (**b**) schematic of the SiHJ photoanode; (**c**) schematic of the SiHJ photocathode.

Study [77] suggests a practical way to improve the photoresponse of Si-based planar photocathodes based on the creation of a Si junction. Here, a soft and feasible strategy for fabricating an amorphous Si/crystalline Si (a-Si/c-Si) junction photocathode with a Pt co-catalyst is proposed. The a-Si layer acts as an electron transfer agent to form a deep a-Si/c-Si junction with a large energy band shift, thereby allowing easy charge transfer. Meanwhile, the a-Si layer serves as a growth substrate to obtain controllable Pt nanoparticles. The Pt/a-Si/c-Si photocathode demonstrates an onset potential of 0.42 V. The photocurrent up to $-35.0$ mA cm$^{-2}$ was obtained on Pt/a-Si/c-Si, which is much higher than without the a-Si layer.

Based on the above, we can conclude that improvements in the production technology of heteroprocessed solar cells and new engineering solutions in the design of photocells also lead to progress in other areas of solid-state engineering and technology.

## 5. Conclusions

In this paper, we briefly talked about the history of the invention and research of the HJT structure solar cells from the studies of the amorphous silicon/crystalline silicon junctions to the novel designs of the HJT solar cells. This work includes the history of development of solar cells, ranging from small-sized heterojunction amorphous/crystalline silicon solar cells research to finished industrial sized solar cells. It also describes the ways that were performed to possibly increase the main output characteristics and technological solutions that affect them. We also raised questions about the challenges of further optimizing HJT solar cell performance by means of numerical simulation programs and current hypotheses. According to the presented information, the authors of this research work propose further ways of the development of heterojunction solar cells. In accordance with the data presented, possibilities were found to increase the output characteristics by improving the design of the contact grid of solar cells and modifying the structure of heterojunction solar cells. According to this work, the most optimal design of the contact grid is the IBS design; due to this design, it is possible to increase the light-absorbing surface of the solar cell and, thereby, to increase the short circuit current of the device. Another possible approach includes the use of a superlattice of quantum wells in the area of the p–i–n-heterojunction as one way performance development. Another way is the use of an attractive new tandem structure based on use of perovskite materials.

Based on the foregoing information, the authors conclude that HJT technology remains promising in terms of the development and implementation of solar energy of this type in the global share of the energy market.

**Author Contributions:** N.C.—Head Leader of grant; concept, management, analysis and writing of papers. K.Z.—Main technologist, growth of thin films. K.A.—electrical characterization of simples. S.Z.—organic solar cells, wet chemistry. A.S.—resources, funding acquisition. All authors have read and agreed to the published version of the manuscript.

**Funding:** This research was funded by Science Committee of the Ministry of Science and Higher Education of the Republic of Kazakhstan, grant number AP09259279.

**Institutional Review Board Statement:** Not applicable.

**Informed Consent Statement:** Not applicable.

**Data Availability Statement:** Not applicable.

**Acknowledgments:** For help in publishing this article, we sincerely thank the entire staff of the Institute of Physics and Technology (Almaty, Kazakhstan). For foundation and inspiration in the field of HJT Solar Cells research we sincerely thank S.Zh. Tokmoldin, N.S. Tokmoldin, and E.I. Terukov.

**Conflicts of Interest:** The authors declare no conflict of interest. The funders had no role in the design of the study; in the collection, analyses, or interpretation of data; in the writing of the manuscript; or in the decision to publish the results.

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
