# Peer review of "Development of Hetero-Junction Silicon Solar Cells with Intrinsic Thin Layer: A Review"

_coatings, doi:10.3390/coatings13040796_

Round 1

Reviewer 1 Report

The author has prepared a review article, "Development of Hetero-junction Silicon Solar Cells with Intrinsic Thin Layer: A Review." However, the article is required a lot of improvement. The abstract is very small and should be a summary of the whole article. Add the essential findings and significance of the work. 

In the Introduction: add the research gap, objectives, and significance of the work.

 The article should be rewritten by adding comparative tables, different HIT solar cells' performance, Degradation behaviors, and sustainability issues.  Add more figures related to the above topics.

Author Response

The author has prepared a review article, "Development of Hetero-junction Silicon Solar Cells with Intrinsic Thin Layer: A Review." However, the article is required a lot of improvement. The abstract is very small and should be a summary of the whole article. Add the essential findings and significance of the work. 

In the Introduction: add the research gap, objectives, and significance of the work. – have done

 The article should be rewritten by adding comparative tables, different HIT solar cells' performance, Degradation behaviors, and sustainability issues.  Add more figures related to the above topics

  • Thanks for the comments. I added what I had time to do. I'm sending you the revised version. If I have more time, I can improve the structure of the article, but right now I'm trying to stick to the deadline.

Reviewer 2 Report

As a review, even a brief, I would have imagined a more comprehensive and much more enlightening article. In short, all parts of the manuscript are really short, inconsistent, and more effort needs to be given. In such a way, I do not recommend this manuscript for publication.

In detail

In the introduction (line 14,15) the authors talk about degradation and low-temperature production technology as a big interest, but almost nothing further is said about it in the whole manuscript.

I suggest a more consistent division of sections. Introduction, Review, Discussion and possible solutions, and Conclusions is pretty much general. For example, the authors talk about passivation, intristic layer, low temperature fabrication, Sanyo company achievements, quantum walls..., why not create a special subsection for that? It will make the work clearer.

Similar and unorganized in the work also appear repetitive phrases such as:

·         line 45: The heterojunction silicon solar cell technology was first invented and patented by Sanyo in the early 1990s.

·         line 25: The first HJT solar cells were developed in the 1990s by Sanyo Company with efficiency of 12%.

·         line 179: HJT technology firstly was developed by Sanyo Ltd. in 1994.

It looks like the date is also different.

Another example:

·         line 197: one of the most important parameters is the quality of the crystalline silicon layer in the HJT structure

·         line 212: One of the most important factors affecting the performance of HJT cell is the material interface between…

·          

What does this statement imply for the reader (line 232)? Please describe.

·         We are interested in engineering solutions in the design of the structure, and not in improving the quality of semiconductor and metallic materials.

I suppose in other words it means that the review focuses on the design, construction and fabrication of the HJT.

line 234: Today, HJT solar cells are the solar energy market leader in terms of "price/quality".

Can the authors be more specific or give some price development?

A table showing the evolution of the effectiveness over time of each team/group/company and references to them would be appropriate. I also miss at least some linkage or commentary on Figure 7, which acts more as a filler to the text.

Figure 1: missing reference.

Line 49–63, 119–139, 153–160: : missing references.

Line 257: The highest efficiency values for HJT solar cells are shown by the IBC contact design.

It would be useful to provide a comparison in the paper.

Figure 4b: What is the relationship of the mentioned IV/PV characteristics to the IBC design? How are these curves different from other semiconductor solar cells?

Incorrect format of some citations, typographically incorrect quantities, minor typos.

In conclusion, review lacks consistency and an effort to go into the depth of the issue. And some of the comments mentioned should be taken at least as a minimum to improve it.

Author Response

  • In the introduction (line 14,15) the authors talk about degradation and low-temperature production technology as a big interest, but almost nothing further is said about it in the whole manuscript.

The use of amorphous silicon makes it possible to carry out technological processes for the production of solar cells at low temperatures not exceeding 200o C  (215 line)

About degradation we added (line 152).

  • Similar and unorganized in the work also appear repetitive phrases such as:

Thank you for comments. I fixed something.

  • line 197: one of the most important parameters is the quality of the crystalline silicon layer in the HJT structure - I was referring to the quality of the overall volume of material

 line 212: One of the most important factors affecting the performance of HJT cell is the material interface between…  -  Here I meant the quality at the border of the heterojunction

  • What does this statement imply for the reader (line 232)? Please describe.

«         We are interested in engineering solutions in the design of the structure, and not in improving the quality of semiconductor and metallic materials…»

Our point is that we are not interested in improving the quality of crystalline silicon or the adhesion of metal contacts, because logically, a material with fewer defects will lead to better conductivity. We wanted to look at engineering solutions, such as a BSF layer or something new that would increase the overall efficiency of the device.

  • line 234: Today, HJT solar cells are the solar energy market leader in terms of "price/quality".

Can the authors be more specific or give some price development?

Yes we can, we have both market analysis and cost analysis of laboratory samples of two different technologies, but we will not, we consider it unnecessary in this section. I decided to delete this line, it is not necessary here.

  • A table showing the evolution of the effectiveness over time of each team/group/company and references to them would be appropriate. I also miss at least some linkage or commentary on Figure 7, which acts more as a filler to the text.

I added information

  • Line 49–63, 119–139, 153–160: : missing references.

These are well-known facts or the links that are there refer to a large piece of text. I gave you a link describing the PECVD spraying process

  • Line 257: The highest efficiency values for HJT solar cells are shown by the IBC contact design.

It would be useful to provide a comparison in the paper.

  • Added description line 455

  • I suggest a more consistent division of sections. Introduction, Review, Discussion and possible solutions, and Conclusions is pretty much general. For example, the authors talk about passivation, intristic layer, low temperature fabrication, Sanyo company achievements, quantum walls..., why not create a special subsection for that? It will make the work clearer.

I think you're right, but I don't see obvious titles for subheadings, and I also don't want to divide the work into small parts, but if I find some solution for that, I'll definitely make changes.

PS: The literature reference numbers are mixed, but we will put everything in order before the final version is published (if approved). 

Reviewer 3 Report

Author/s reported the brief history of heterojunction Si-Solar cell. Presenting history is fine, but it should have some meaningful outcome for scholars who are planning or just start working in this area. I am suggesting the author add the following 

a) Most of the references are before 2015. Authors should add the latest research work with the latest citation.

b) Author should add an HJ Si-solar cell comparison table of progress done by researchers. I found no comparison table for efficiency progress for the different device structures. Because the comparison table gives a lot of information in a very short way. 

c) If authors feel Ok, I am suggesting adding more tandem structures also. That will be helpful for newcomers in this area.   

Author Response

Author/s reported the brief history of heterojunction Si-Solar cell. Presenting history is fine, but it should have some meaningful outcome for scholars who are planning or just start working in this area. I am suggesting the author add the following 

a) Most of the references are before 2015. Authors should add the latest research work with the latest citation. - it will be difficult to redo the whole review with newer works, but I will add a section with modeling with links to the newest works

b) Author should add an HJ Si-solar cell comparison table of progress done by researchers. I found no comparison table for efficiency progress for the different device structures. Because the comparison table gives a lot of information in a very short way. – Figure 9 with description and references to Green`s Solar Tables.

c) If authors feel Ok, I am suggesting adding more tandem structures also. That will be helpful for newcomers in this area. – We did it (figure 10 and 11 with description)

PS: The literature reference numbers are mixed, but we will put everything in order before the final version is published (if approved). 

Round 2

Reviewer 2 Report

Although the manuscript quality has been relatively improved it has not helped the overall clarity and still contains some flaws. I strongly recommend using passive voice. Not all my questions have been addressed either.

I don't follow the authors statement if I find some solution for that, I'll definitely make changes. Once the manuscript is submitted, the changes are considered to be finished.

I understand that the authors don't want to interfere too much with the flow of the text, but sections like Review are rather uninformative and may even lower their own future citability. After all, the whole paper is a review.

Author Response

Thanks for your comments and tips! You certainly helped us make the manuscript better! An outside perspective is very important.

We have worked on the structure of the article and made some additions as well. 

Unfortunately, I did not understand what your questions were left unanswered. 

Reviewer 3 Report

fine now. 

Author Response

Thank you so much! It was a pleasure to hear your approval. We added a little more to the manuscript and worked on its structure. 
